# Bolt-Loosening Detection Using 1D and 2D Input Data Based on Two-Stream Convolutional Neural Networks

**DOI:** 10.3390/ma15196757

**Published:** 2022-09-29

**Authors:** Xiaoli Hou, Weichao Guo, Shengjie Ren, Yan Li, Yue Si, Lizheng Su

**Affiliations:** 1School of Mechanical and Precision Instrument Engineering, Xi’an University of Technology, Xi’an 710048, China; 2Xi’an Institute of Electronic Engineering, Xi’an 710100, China

**Keywords:** bolt connection, fault diagnosis, two-stream convolutional neural networks, anti-noise

## Abstract

At present, the detection accuracy of bolt-loosening diagnoses is still not high. In order to improve the detection accuracy, this paper proposes a fault diagnosis model based on the TSCNN model, which can simultaneously extract fault features from vibration signals and time-frequency images and can precisely detect the bolt-loosening states. In this paper, the LeNet-5 network is improved by adjusting the size and number of the convolution kernels, introducing the dropout operation, and building a two-dimensional convolutional neural network (2DCNN) model. Combining the advantages of a one-dimensional convolutional neural network (1DCNN) with wide first-layer kernels to suppress high-frequency noise, a two-stream convolutional neural network (TSCNN) is proposed based on 1D and 2D input data. The proposed model uses raw vibration signals and time-frequency images as input and automatically extracts sensitive features and representative information. Finally, the effectiveness and superiority of the proposed approach are verified by practical experiments that are carried out on a machine tool guideway. The experimental results show that the proposed approach can effectively achieve end-to-end bolt-loosening fault diagnoses, with an average recognition accuracy of 99.58%. In addition, the method can easily achieve over 93% accuracy when the SNR is over 0 dB without any denoising preprocessing. The results show that the proposed approach not only achieves high classification accuracy but also has good noise immunity.

## 1. Introduction

Bolted connections are frequently and widely applied in mechanical equipment because a bolted connection has remarkable advantages, e.g., low cost, high reliability, and easy installation and replacement. However, bolt loosening often occurs due to the fact that the mechanical equipment usually operates under time-varying loads, and bolt loosening normally has a negative impact on the normal operation of equipment such as poor equipment accuracy, significant economic losses, and even accidents. Therefore, developing an effective method for bolt-loosening level recognition is an important task for preventing any accidents in the normal operation of mechanical equipment [1,2]. In the past few decades, the detection of bolt loosening has usually been performed using vibration signals [2,3]. In order to establish the relationship between the mechanical structure and loosening vibration signal, some researchers have extracted time-domain and frequency-domain features and then used various classifiers to detect bolt loosening according to the extracted fault features. The common classifiers for fault diagnosis include k-nearest neighbor (KNN), support vector machine (SVM), k-means clustering, and so on. However, during this kind of detection process, it has been noted that hand-crafted features may miss some distinguishing features about bolt loosening, which means that the classification results are not accurate or reliable [4,5]. For this reason, a method that can automatically extract features is highly sought after.

Fortunately, in recent years, deep learning has developed rapidly and has been successfully applied in speech recognition [6], face recognition [7], computer vision [8], fault diagnosis [9], and other fields because the advantage of deep learning is that it uses unsupervised or semi-supervised feature learning and hierarchical feature extraction algorithms instead of manual feature extraction. Furthermore, another remarkable advantage of deep learning is that it uses multiple-layer artificial neural networks with representation learning, which can extract higher-level features from the raw input data. Simonyan et al. [10] proposed a two-stream convolutional network architecture that incorporates spatial and temporal networks. Guo et al. [11] proposed a novel diagnosis method using a convolutional neural network (CNN) to directly classify the continuous wavelet transform scalogram (CWTS). Wen et al. [12] adopted a new CNN based on LeNet-5 for fault diagnosis, which converts one-dimensional (1D) signals into two-dimensional (2D) images. Although every artificial intelligent method can miss features, deep learning usually automatically extracts features from 2D images and avoids extracting features that heavily rely on human intervention. In the above researches, it was found that scholars mostly converted the original signal into grayscale images or time–frequency images, and then deep learning was used for the fault diagnosis. However, the vibration, pressure, and sound signals measured by the sensors were all one-dimensional time signals in the fault diagnosis. Time-series signals usually contain important features about equipment conditions, such as periodic and short-term pulse features, which may not exist in two-dimensional images. Therefore, one-dimensional time signals are widely used in many fields and produce some remarkable results. For example, in real engineering fields, Yan et al. [13] applied a one-dimensional convolutional neural network (1DCNN) to detect chiller faults and Xiang et al. [14] carried out a study on wind turbine fault detection using 1DCNN and SCADA data analysis. Moreover, Acharya et al. [15] applied 1DCNN to help clinicians detect ECG signals for distinguishing between healthy individuals and those with myocardial infarction. Abdeljaber et al. [16] presented a novel, fast, and accurate structural damage detection system using 1DCNN that had an inherent adaptive design to detect the structural damage of the stand simulator. Peng et al. [17] used 1DCNN to diagnose the vibrations of wheel bearings of high-speed trains and achieved reasonable results. Zhang et al. [18] proposed a novel method called a deep convolutional neural network with wide first-layer kernels (WDCNN), which used raw vibration signals as input and had wide kernels in the first convolutional layer for suppressing high-frequency noise. This novel method showed a good classification ability under different working loads and noise environments. From the above applications, it can be seen that 1DCNN has a strong classification ability in real-time fault diagnosis. 

Moreover, a few experts have also applied machine learning methods to diagnose bolt-loosening faults. Kong et al. [3] provided a percussion method based on the power spectral density (PSD) of the sound signal and the decision tree (DT) to identify the bolt-loosening level. Zhang et al. [19] extracted the Mel-frequency cepstral coefficients (MFCCs) of the sound signal as frequency-domain feature parameters, and a support vector machine classifier was trained to effectively detect bolt loosening. 

However, at present, there are still many difficulties in the diagnosis of bolt loosening. For instance, the extraction of time-domain and frequency-domain features are still highly dependent on manual intervention. Considering that the manual intervention of feature extraction significantly affects the identification accuracy, it is difficult to extract the bolt-loosening features and construct the loosening feature set. Therefore, the entire process is usually time-consuming for bolt fault diagnoses, to say nothing of automatic detection. Furthermore, in bolt-loosening fault diagnoses, problems such as low model recognition accuracy and effectivity in noisy environments cannot be ignored.

In order to address these problems, a new fault diagnosis method for bolt loosening is proposed based on two-stream convolutional neural networks (TSCNN) simultaneously using one-dimensional signals and two-dimensional images. The feasibility and effectiveness of the method are verified via experimental data. The remainder of this paper is organized as follows: Section 2 introduces the basic theories of convolution, activation, and pooling, with particular attention to the difference between 1DCNN and 2DCNN. Then, the specific network structure and diagnostic procedure of the TSCNN model are built in Section 3. Section 4 describes the experimental setup, and the effectiveness of the proposed method is demonstrated by comparing it with other diagnostic methods in different noise environments. Finally, some conclusions are drawn in Section 5.

## 2. Theoretical Background

Convolutional neural networks have fewer parameters and directly use original image data as input data. Thus, manual participation is not required in preprocessing, which means that CNN can be widely applied in image processing. A CNN usually consists of convolution layers, pooling layers, and fully connected layers. According to model structures and performance algorithms, LeNet, ResNet, AlexNet, and GoogleNet are the most popular CNN networks. Compared to the other types of networks, LeNet is applied in many real engineering fields because it has a simple model structure and fewer setting parameters. For instance, Table 1 shows the structure settings for the LeNet-5 network, which is a typical CNN. The classical LeNet-5 network consists of two convolution layers (i.e., Conv1 and Conv2), two pooling layers (i.e., Pooling1 and Pooling2), and three fully connected layers (i.e., FC1, FC2, and FC3) [18].

### 2.1. Convolutional Layer

Convolution was applied to check the input local area for convolution operations and generate corresponding features. The most important feature of this layer is weight sharing, which greatly reduces the network parameters of the convolutional layer. If the *l*-th layer is assumed as a convolutional layer, its convolution process can be described as follows [18]:(1)$$y_{j}^{l+1}=f(\sum_{i=1}^{M}\left(x_{i}^{l}⊗k_{ij}^{l}\right)+b_{j}^{l}),$$
where the notation ⊗ represents the convolution operation; $y_{j}^{l+1}$ is denoted as the input of the *j*-th neuron at layer *l* + 1; *f* represents the activation functions; $x_{i}^{l}$ is the output of the *i*-th neuron at layer *l*; $k_{ij}^{l}$ is defined as the kernel from the *i*-th neuron at layer *l* to the *j*-th neuron at layer *l* + 1; *M* denotes the number of feature maps, which is same as the number of the kernel at layer *l*; and $b_{j}^{l}$ is the bias of the *j*-th neuron at layer *l* + 1.

Furthermore, in this paper, the Rectified Linear Unit (ReLU) was employed as the activation layer to enhance the divisibility of the extracted features via a nonlinear operation. ReLU has good sparsity compared to the traditional activation functions, Sigmoid and Tanh functions, and more than 50% of neurons in the ReLU are activated. Moreover, the ReLU only needs to add, multiply, and compare operations, which makes calculations efficient and fast. In terms of optimization, the ReLU is a left-saturated function, which can effectively alleviate the problem of gradient disappearance [20]. The ReLU can be presented as
(2)$$a_{j}^{l+1}=f\left(y_{j}^{l+1}\right)=\max\left(0,y_{j}^{l+1}\right)=\left\{\begin{array}{l} \;y_{j}^{l+1},\;\;\;\;\;if\;y_{j}^{l+1}\geq 0\\ \;0,\;\;\;\;\;\;\;\;else \end{array}\right.\;,$$
where $a_{j}^{l+1}$ is the activation of $y_{j}^{l+1}$.

### 2.2. Pooling Layer

The pooling layer in the CNN compresses the feature map. This means that the pooling layer not only extracts the main features but also reduces the parameters of the network. The pooling layer can be divided into max pooling, average pooling, and weighted pooling. The most commonly used pooling layer is the max pooling layer, which is defined as
(3)$$P_{j}^{l+1}(i)=\underset{(i-1)W+1\leq t\leq iW}{\max}\left\{\left.a_{j}^{l}(t)\right\}\right.,$$
where $P_{j}^{l+1}(i)$ denotes the corresponding value of the neuron in layer *l* + 1 of the pooling operation and *W* denotes the width of the pooling region.

### 2.3. Fully Connected Layer

The function of a fully connected (FC) layer is to fully connect all the neurons. Normally, the FC layer is also linked to the output layer and performs as a classifier [21]. The classifier employs a supervised learning algorithm to solve multi-classification problems. The output value is described as follows:(4)$$z_{j}^{l+1}=\sum_{i=1}^{n}w_{ij}^{l}P_{j}^{l}+b_{j}^{l},$$
where $z_{j}^{l+1}$ represents the output value of the *j*-th neuron at layer *l* + 1 and $w_{ij}^{l}$ is defined as the weight between the *i*-th neuron at layer *l* and the *j*-th neuron at layer *l*+1.

In the output layer, the softmax function is applied to transform the logits of the neuron into the form of the probability distribution. The normalized probability *q*(*z_j_*) obtained by the softmax function is described as
(5)$$q(z_{j})=\mathrm{softmax}\left(z_{j}\right)=\frac{\exp(z_{j})}{\sum_{n=1}^{N}\exp(z_{j})},$$
where *z_j_* denotes the logits of the *j*-th output neuron and *N* is the total number of classes.

### 2.4. Filter Sliding for 1DCNN and 2DCNN

The input of the 1DCNN is a one-dimensional time series and the data are the vibration signals recorded by the dynamical sensors without any treatment. This guarantees the authenticity of the signals. Moreover, only one-dimensional convolution is employed with fewer parameters, which can significantly reduce computing resources and time. A 1DCNN is frequently applied in natural language processing, sequence models, etc. Remarkably, a 2DCNN performs very well for extracting local features of images, thus it frequently performs in image processing, computer vision, and the like.

The difference between a 1DCNN and a 2DCNN is that the input data dimensions and filter sliding methods are different. As shown in Figure 1, the row of a 1DCNN represents the vibration signal of the measuring point and the column represents the sensor or channel such as the original signal in the x, y, or z direction. The filter has only 1 degree of freedom to slide along the time series. A one-dimensional filter requires that the width and number of the channels of the filter and data are equal, covering all columns at any time, and its height determines the length of each convolution operation. However, 2DCNN filters need to slide horizontally, and then vertically move to the next location to continue a horizontal sliding through the entire image. Thus, there are two degrees of freedom, which requires that the number of channels of the filter and data are equal, and its height and width determine the range of each convolution operation. In addition, a 1DCNN and a 2DCNN almost have the same architecture and working principles [22].

## 3. The Proposed WDCNN Bolt-Loosening Fault Diagnosis Model

Usually, one kind of CNN only permits one kind of input data. However, as mentioned previously, simultaneously using 1D time-varying signals and 2D image data could potentially improve the accuracy of diagnosing bolt-loosening faults. Thus, the frameworks of the 1DCNN and 2DCNN are improved in this section. Afterward, an improved diagnosis fault model for bolt loosening is proposed using the TSCNN.

### 3.1. The Structure and Parameter Settings Based on an Improved 1DCNN

As shown in Table 2, the improved 1DCNN included five convolutional layers and five pooling layers. Wide kernels in the first convolutional layer (64 × 1) can better suppress high-frequency noise [18], and the multi-layer small convolution kernels (3 × 1) make the network deeper, which helps multi-layer nonlinear mapping and improves network performance [23]. Moreover, the size and number of kernels significantly affect the output accuracy of the network so researchers have carried out many studies by adjusting the kernel settings [24,25].

The pooling layers performed a 2 × 1 max pooling operation after using the ReLU instead of the Sigmoid activation function. Afterward, a two-stream fusion of the fault information extracted by the 1DCNN and 2DCNN was performed. The FC1 layer was a full-connection layer with 120 neurons, which was fully connected to the output of the two-stream fusion layer and produced 120 feature maps of 1 × 1 pixels. The FC2 layer was a full-connection layer with 84 neurons, which calculated the dot-product between the input vector and weight vector and added the bias value, and the results were outputted by the ReLU function. The FC3 layer was the output layer, which had six neurons and divided all the input images into six different categories corresponding to the bolts under six different loosening conditions; the loosening conditions can be seen in the table in Section 4.1 The dropout operation was performed after the FC1 and FC2 layers, and its value was set to 0.2 [26], which effectively prevented overfitting during learning the fault diagnosis model. In addition, using the ReLU function achieved a full backpropagation calculation without causing gradient disappearance.

### 3.2. The Structure and Parameter Settings Based on an Improved 2DCNN

Before fault diagnosis, the vibration signals needed to be converted into images to match the 2DCNN input. The short-time Fourier transform (STFT), also known as the window Fourier transform, is an extensive and effective time–frequency analysis method. The basic principle is to assume that the signal is stable for a short period, and the time-domain signal can be divided into a limited number of small segments. Afterward, the time-varying frequency spectrum of the entire signal is produced by continuously sliding the time window to the end of the signal via Fourier transform [27]. As shown in Figure 2, the original signal was divided into multiple data samples with oversampling, and *STFT* was used to obtain the time–frequency characterization of each sample, which was beneficial to the generalization ability of the model. The *STFT* can be presented as
(6)$$STFT(t,\omega )=\int_{-\infty }^{\infty }x(\tau )w(\tau -t)e^{-j\omega t}d\tau ,$$
where *t* is the shift factor and τ is the time position center of the window function w(t).

The network structure of the improved 2DCNN included four convolution layers and four pooling layers, also using the ReLU activation function and dropout operation. In the training of the bolt-loosening fault diagnosis model, it was found that with the increase in the image size, the diagnosis accuracy increased but the training speed became slower. Considering the fault diagnosis accuracy and training speed, the final time–frequency image size was set to 64 × 64. The detailed settings are listed in Table 3.

Compared with the traditional LeNet-5 network, the improved 2DCNN and 1DCNN mainly had the following five advantages.

(1)The size of the input layer was changed. The improved 1DCNN inputs were the original vibration signals of 4096 × 1, and the improved 2DCNN inputs were the time–frequency images with a size of 64 × 64.(2)The number of convolutional layers and pooling layers was increased. Theoretically, the deeper the neural network, the stronger the feature expression ability, However, more convolutional layers and pooling layers easily lead to network degradation. Finally, five convolutional layers and five pooling layers were used in the improved 1DCNN, and four convolution layers and four pooling layers were used in the improved 2DCNN.(3)The size and number of the convolution kernels were changed. Due to the complexity of the vibration signals, the size and number of the convolution kernels needed to be adjusted to enhance the network’s classification ability. The adjusted convolution kernel settings are shown in Table 2 and Table 3.(4)The dropout operation was used, which effectively reduced overfitting during the training of the fault diagnosis model.(5)The ReLU activation function was adopted. The ReLU activation function prevented gradient disappearance and the training speed was faster than the Sigmoid activation function.

### 3.3. The Bolt Diagnosis Model Based on the TSCNN

Figure 3 shows the fault diagnosis process for bolt loosening based on the TSCNN model. As shown in Figure 3, the inputs of this network were a vibration signal and time–frequency image, the features extracted by the 1DCNN and 2DCNN were fused with the Softmax classification, and then the bolt-loosening fault diagnosis was performed. The diagnosis process is described as follows:

Step 1: The vibration signals of bolt loosening were collected by the sensors deployed on the guideway of the machine tool. The training samples were dealt with by oversampling in order to reduce the effect of the overfitting problem.

Step 2: The preprocessed vibration signals in Step 1 were converted into time–frequency images with STFT, and the sizes of the images were set to 64 × 64.

Step 3: The original vibration signal was divided into training set 1 and test set 1 with a ratio of 8:2. At the same time, the time–frequency images were also divided into training set 2 and test set 2 with the same ratio.

Step 4: The bolt-loosening fault diagnosis model was trained using training sets 1 and 2.

Step 5: The bolt-loosening fault diagnosis model was employed to identify the test set online. The validity of the diagnosis model was evaluated via the identified results, and the network parameters were optimized in order to obtain a reliable and effective bolt-loosening fault diagnosis model.

Step 6: Real bolt conditions were recognized by the optimized bolt-loosening fault diagnosis model and some equipment maintenance advice was provided according to the recognition results.

### 3.4. Model Evaluation Index

After the bolt loosening, the fault diagnosis model was well trained, and the recognition accuracies and loss value indexes were observed to evaluate the quality of the diagnosis model. The accuracy rate represents the ratio of the number of correct predictions to the total number. The loss function of the model is the cross-entropy between the estimated softmax output probability distribution and the target class probability distribution. The *Loss* value was smaller and the accuracy of the diagnostic model was higher. The formula is as follows:(7)$$Loss=-\sum_{x}p(x)\log\;q\left(x\right),$$
where *p*(*x*) denotes the target distribution and *q*(*x*) denotes the estimated distribution.

## 4. Experiment and Analysis

The TSCNN model was implemented in MATLAB 2020a and Pytorch 1.5.1. The TSCNN model was trained on a Windows 10 computer with a Hexa-core Intel i5-8400H CPU at 2.8 GHz and 16 GB RAM. All experiments were conducted using adaptive moment estimation (Adam), where the initial learning rate was set to 0.008 by comprehensively considering the classification accuracy and convergence speed. It was important to select the right batch size during the training and testing. According to our previous work, both batch sizes were set to 128.

### 4.1. Experimental Setup

The experiment for recognizing bolt loosening under the same working conditions was carried out on a machine tool guideway (size: 1500 mm × 130 mm × 50 mm), which is illustrated in Figure 4. Twelve Q235 T-bolts were arranged on both sides of the guideway. Different loosening conditions in the six bolts were analyzed due to the symmetry of the bolt layout on the guideway. For convenience, the six bolts were numbered and the details can be seen in Figure 4. Different torques of bolts were set via a digital torque wrench (SATA 96525) to simulate different loosening failures. One acceleration sensor (PCB 333B30) was glued to the guideway and connected to the data acquisition instrument (LMS SCADAS Mobile) through connection line to collect vibration data of different cases with a sampling rate of 12.8 kHz. The sensitivity of the acceleration sensor was 103.4 mv/g. The LMS data acquisition instrument was connected to the laptop by the network cable. The standard torque for the grade 8.8 M12 steel bolt was 80 Nm. Therefore, the bolts were torqued to 80 Nm by a digital torque wrench. The bolts in this condition were considered to be in a healthy state without any loosening.

Table 4 shows that six kinds of bolt looseness were set in this experiment, such as T1 (all bolts tightened with torques of 80 Nm), T2 (only bolt 1 was severely loose with a torque of 0 Nm), T3 (bolt 1 and 2 were severely loose with torques of 0 Nm), T4 (all bolts were slightly loose with torques of 60 Nm), T5 (all bolts were moderately loose with torques of 40 Nm), and T6 (all bolts were severely loose with torques of 0 Nm). At each torque level, five assemble–disassemble–repeat tests were performed to validate the robustness and repeatability of the proposed method. Each sample contained 4096 points and the details of all the datasets are also depicted in Table 4.

### 4.2. Training and Verification of the Bolt Fault Diagnosis Model

In order to demonstrate the effectiveness of the proposed method in this paper, the fault diagnosis model based on the TSCNN model was trained and tested with the dataset listed in Table 4. The total computing time was about 900 s and ran on the laptop mentioned in the previous section. The accuracy and loss curves of the bolt fault diagnosis model obtained during the training and test phases are shown in Figure 5 and Figure 6, respectively.

As shown in Figure 5, during the training and test phases of the TSCNN fault diagnosis model, it can be seen that the initial diagnosis accuracy was continuously improved, and the fault classification accuracy stabilized after about 70 iterations. When the iteration reached 100, the diagnosis accuracy could reach up to 99.58%. On the other hand, Figure 6 shows that the loss function value decreased rapidly in the first 70 iterations, but the loss function values of the test dataset were much high than the values for the training dataset. This is because the extracted fault features for the test dataset and the training dataset were not exactly the same. However, with the evolution of the TSCNN model, the loss function curves for the test dataset and training dataset decreased rapidly and both curves tended to zero after 70 iterations. This indicates that the TSCNN model had high accuracy, fast convergence speed, and no overfitting.

Figure 7 clearly shows the classification effect of the fault diagnostic model on the test set. The x-axis represents 240 sets of test data and the y-axis represents six types of faults. It can be seen that most test sets were classified correctly except for T2. One of the test samples T2 was wrongly grouped with T4, i.e., the loosening of bolt 1 was incorrectly classified as the loosening of bolts 1 and 2. Thus, the diagnostic accuracy of the fault diagnostic model was 97.5% in recognizing T2 faults.

In order to verify the diagnosis performance of the proposed method, we compared it with the CNN-SVM and AlexNet algorithms.

Table 5 describes the classification accuracy of bolt loosening under different diagnosis models. As seen in Table 5, for the CNN-SVM diagnosis model, the diagnosis accuracies of the six failure types were 92.5%, 95%, 97.5%, 75%, 100%, and 90%, respectively. For the fine-tuned AlexNet diagnosis model, the recognition accuracies were 100%, 90%, 92.5%, 90%, 95%, and 95%, respectively. For the TSCNN diagnosis model, the recognition accuracies reached 100%, 97.5%, 100%, 100%, 100%, and 100%, respectively.

Although the AlexNet and LeNet networks had similar network structures, compared to LeNet, AlexNet utilized the GPU to treat the matrix operation, which accelerated the CNN model training. Hence, AlexNet had obvious advantages in dealing with a large data problem. AlexNet also performed well in a large and deep network. This means that AlexNet could extract more kinds of features and had a good generalization ability. However, it could see that the average accuracy of the TSCNN diagnosis model was the highest because the average accuracies for the three fault diagnosis models were 91.67%, 93.75%, and 99.58%, respectively. Therefore, it can be said that the proposed TSCNN was more effective for diagnosing the different loosening fault states for bolt connections compared to the other two models.

### 4.3. Performance under Noise Environment

The working conditions of the bolt structure are accompanied by varying degrees of noise in actual industrial applications. In order to further verify the applicability of the proposed TSCNN model, we added white Gaussian noise to the original vibration signals in the following cases and introduced a signal-to-noise ratio (*SNR*_dB_) to evaluate the level of white Gaussian noise. The definition of *SNR*_dB_ is described as follows:(8)$$SNR_{\mathrm{dB}}=10\log_{10}\left(\frac{P_{\mathrm{signal}\;}}{P_{\mathrm{noise}\;}}\right)=20\log_{10}\left(\frac{A_{\mathrm{signal}\;}}{A_{\mathrm{noise}\;}}\right),$$
where the subscript dB means that the signal-to-noise ratio is expressed in decibels. $P_{\mathrm{signal}\;}$ and $A_{\mathrm{signal}}$ are the power and amplitude of the signal and $P_{\mathrm{noise}}$ and $A_{\mathrm{noise}}$ are the power and amplitude of the noise, respectively.

Figure 8 shows that the original signal of the bolt 2 loosening fault was added with the white Gaussian noise. The SNR for the composite noisy signal was 0 dB, which means that the power of the noise was equal to that of the original signal. To verify the anti-noise ability of the proposed model, we tested the CNN-SVM, AlexNet, and proposed TSCNN models with five different SNR levels, i.e., SNRs in the range of −4 dB, −2 dB, 0 dB, 2 dB, and 4 dB.

The fault diagnosis results of the three diagnosis models in noisy environments with different SNR levels are shown in Figure 9. It can be seen that following the increase in the SNR, the recognition accuracies of the three diagnosis models also increased. It should be noted that the recognition accuracies of the TSCNN model were always above 90% for the different SNR levels. Even at the highest noise level, for instance, SNR = −4, the recognition accuracy of the TSCNN model was 92%. At the same SNR level, the diagnosis accuracy of the CNN-SVM and AlexNet models was less than 50%. Although the SNR was larger than 0 dB, the recognition accuracy of the TSCNN model reached 98.75%. In contrast, the fault diagnosis accuracies of the CNN-SVM and AlexNet models were still below 85%. Hence, it can be seen that the TSCNN model performed better than the CNN-SVM and AlexNet models for bolt-loosening diagnoses even with high noise.

## 5. Conclusions

This paper proposes a new fault diagnosis model based on the TSCNN model to address the bolt-loosening fault diagnosis problem. The TSCNN model simultaneously extracts fault features from vibration signals and time–frequency images to detect the bolt connection state. The diagnosis ability of the model is verified by the experiment and is compared to the other two fault diagnosis models. The detailed conclusions are as follows:(1)The dynamic characteristics of the structure change with the loosening of the bolts on the machine tool guideways and the vibration signal and its frequency spectrum also change. In this paper, the TSCNN model is developed to simultaneously integrate the fault features extracted by the 1DCNN and 2DCNN, which can accurately represent bolt-loosening situations.(2)The CNN is improved by applying wide kernels in the first convolutional layer (64 × 1) and successive small convolution kernels (3 × 1). Wide kernels can suppress high-frequency noise and successive small convolution kernels lead to deeper multilayer nonlinear mapping, which enhances the fault classification capability.(3)Adopting the ReLU activation function and performing the dropout operation can significantly improve the convergence speed and generalization ability of the model.(4)The TSCNN model effectively avoids manual feature extraction and low-efficiency post-processing. The ability to recognize different degrees of bolt-loosening situations is assessed via real experiments and compared to the other two fault diagnosis models. The results show that the average recognition accuracies of the TSCNN model can reach 99.58% and easily achieve over 93% once the SNR is over 0 dB without any denoising preprocessing. Hence, it can be said that the TSCNN model can achieve high classification accuracy for bolt-loosening fault diagnoses and is also very effective and robust even in a high-noise environment.

## Figures and Tables

**Figure 1 materials-15-06757-f001:**
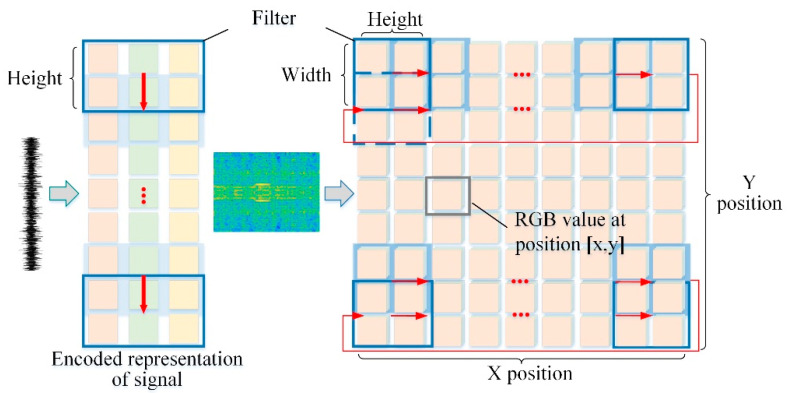
The sliding direction of the filter on a 1DCNN and 2DCNN, respectively.

**Figure 2 materials-15-06757-f002:**
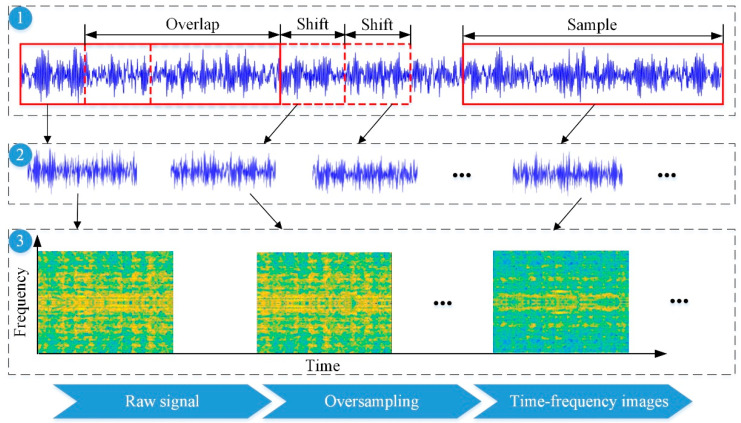
Signal preprocessing process.

**Figure 3 materials-15-06757-f003:**
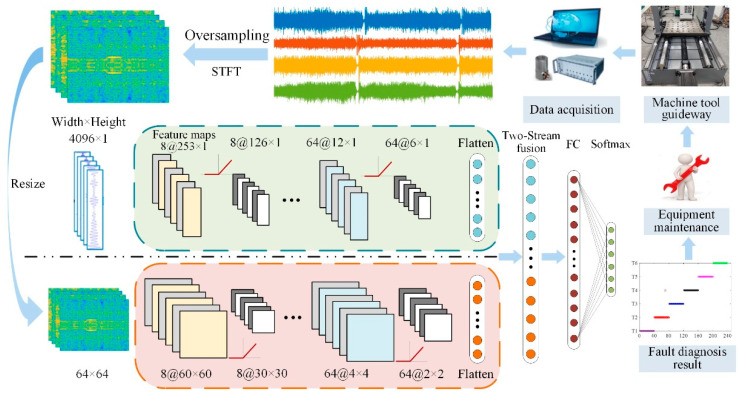
The structure illustration of the TSCNN model.

**Figure 4 materials-15-06757-f004:**
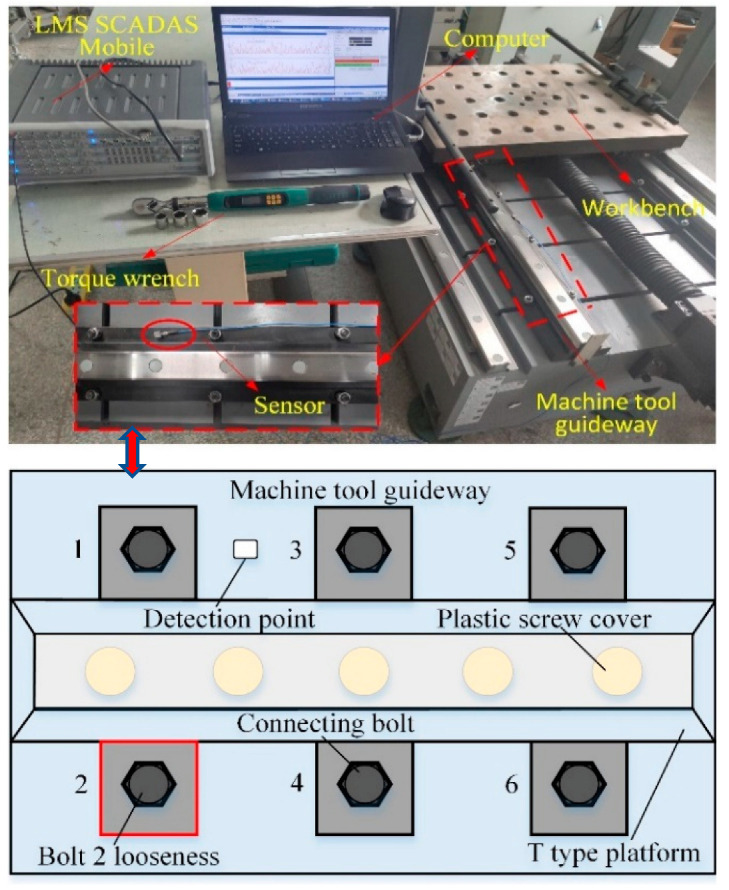
Experimental setup for bolt loosening.

**Figure 5 materials-15-06757-f005:**
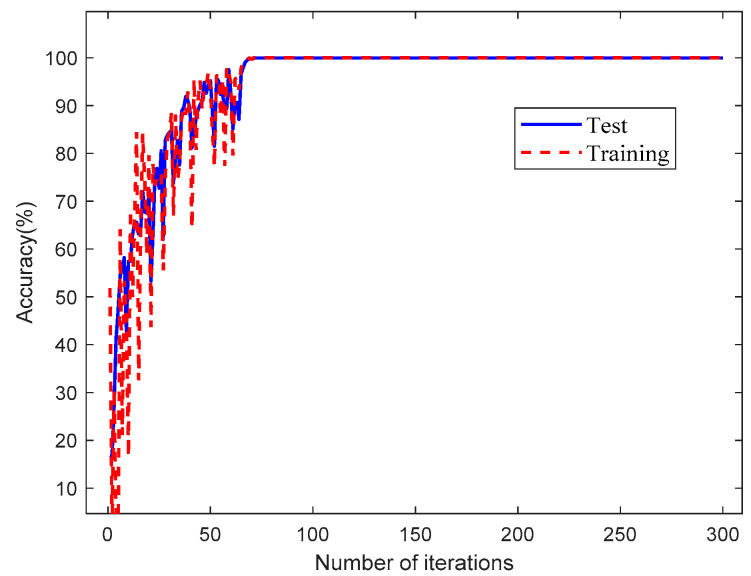
Accuracy curve of the fault diagnosis model based on the TSCNN.

**Figure 6 materials-15-06757-f006:**
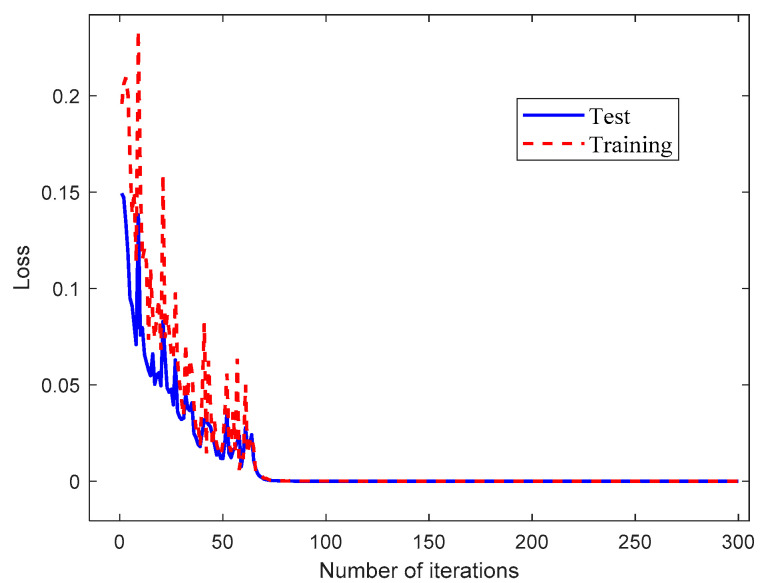
Loss function curve of the fault diagnosis model based on the TSCNN.

**Figure 7 materials-15-06757-f007:**
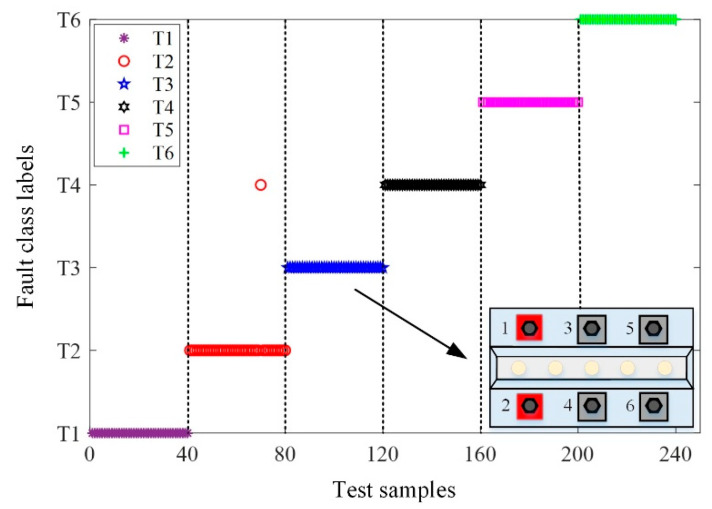
Prediction results on the test set.

**Figure 8 materials-15-06757-f008:**
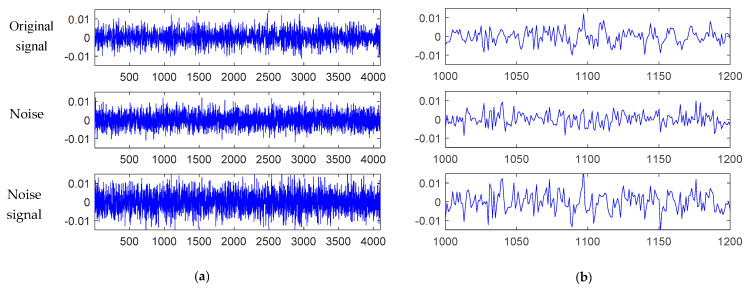
Illustration of the original signal of bolt 2 loosening fault with white Gaussian noise. The composite noise signal with SNR = 0 dB. (**a**) Original signal, Gaussian noise, and noise signal; (**b**) Zoom in the three signals to the range [1000, 1200].

**Figure 9 materials-15-06757-f009:**
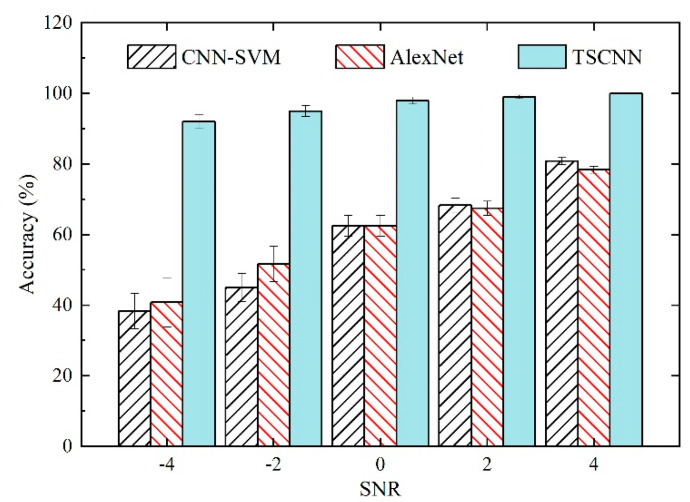
Diagnosis accuracies of the three diagnosis models in different noise environments.

**Table 1 materials-15-06757-t001:** Detailed settings of LeNet-5 network.

Layer Type	Kernel Size/Stride	Kernel Number	Output Size	Activation Function
Input			32 × 32	
Conv1	5 × 5/1	6	6@28 × 28	Sigmoid
Pooling1	2 × 2/2	6	6@14 × 14	
Conv2	5 × 5/1	16	16@10 × 10	Sigmoid
Pooling2	2 × 2/2	16	16@5 × 5	
FC1	120	1	120 × 1	Sigmoid
FC2	84	1	84 × 1	Sigmoid
FC3	10	1	10	

**Table 2 materials-15-06757-t002:** Detailed settings of the 1DCNN.

Layer Type	Kernel Size/Stride	Kernel Number	Output Size	Activation Function
Input			4096 × 1	
Conv1	64 × 1/16	8	8@253 × 1	ReLU
Pooling1	2 × 1/2	8	8@126 × 1	
Conv2	3 × 1/1	16	16@124 × 1	ReLU
Pooling2	2 × 1/2	16	16@62 × 1	
Conv3	3 × 1/1	32	32@60 × 1	ReLU
Pooling3	2 × 1/2	32	32@30 × 1	
Conv4	3 × 1/1	64	64@28 × 1	ReLU
Pooling4	2 × 1/2	64	64@14 × 1	
Conv5	3 × 1/1	64	64@12 × 1	ReLU
Pooling5	2 × 1/2	64	64@6 × 1	
FC1	120	1	120 × 1	ReLU
FC2	84	1	84 × 1	ReLU
FC3	6	1	6	

**Table 3 materials-15-06757-t003:** Detailed settings of the 2DCNN.

Layer Type	Kernel Size/Stride	Kernel Number	Output Size	Activation Function
Input			64 × 64	
Conv1	5 × 5/1	8	8@60 × 60	ReLU
Pooling1	2 × 2/2	8	8@30 × 30	
Conv2	3 × 3/1	16	16@28 × 28	ReLU
Pooling2	2 × 2/2	16	16@14 × 14	
Conv3	3 × 3/1	32	32@12 × 12	ReLU
Pooling3	2 × 2/2	32	32@6 × 6	
Conv4	3 × 3/1	64	64@4 × 4	ReLU
Pooling4	2 × 2/2	64	64@2 × 2	
FC1	120	1	120 × 1	ReLU
FC2	84	1	84 × 1	ReLU
FC3	6	1	6	

**Table 4 materials-15-06757-t004:** Detailed arrangement of experimental cases.

Case	Looseness Extent	Torque (Nm)	Training/Test Dataset
T1	All bolts tightened	80	160/40
T2	Bolt 1 severely loose	0	160/40
T3	Bolt 1 and 2 severely loose	0	160/40
T4	All bolts slightly loose	60	160/40
T5	All bolts moderately loose	40	160/40
T6	All bolts severely loose	0	160/40

**Table 5 materials-15-06757-t005:** Results of different bolt-loosening faults based using the proposed TSCNN compared with CNN-SVM and AlexNet.

Fault Diagnosis Model	Diagnosis Accuracy (%)						
	T1	T2	T3	T4	T5	T6	Average
CNN-SVM	92.5	95	97.5	75	100	90	91.67
AlexNet	100	90	92.5	90	95	95	93.75
TSCNN	100	97.5	100	100	100	100	99.58

## Data Availability

Data are contained within the article. In addition, the data presented are available on request from the corresponding author.

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
