# Peer review of "Bolt-Loosening Detection Using 1D and 2D Input Data Based on Two-Stream Convolutional Neural Networks"

_materials, 2022, doi:10.3390/ma15196757_

Round 1
Reviewer 1 Report
Please revise the manuscript according to the attached file.

Reviewer 2 Report
The authors should emphasize the reasons to choose the used class of convolutional neural networks among other alternatives of the same nature with technical motivations (accuracy, training time, processing time, class of information to process, etc). The presented information is oriented to the selected network (Table 1), but it does not mention a selection process concerning the alternatives of convolutional neural networks (as the authors provided detailed reasons for the improvements with respect to the LeNet5).
The quality of some figures is poor. They have to be improved (Figures 1, 2, 5, 6, 8). If possible, try to use vector formats in their creation to avoid distortion effects.
Figure 8 should include a zoom in a certain part to show in a clearer manner the noise effects. As shown, it is difficult to notice specific abrupt changes, especially in the noise peaks.
The availability of the experimental data for the diagnosis should be a good option for other researchers. There is a specific MDPI journal (Data) that can be an option for a future publication if the authors consider the suggestion.
Was an analysis of false positives/negatives carried out? It can be very helpful for fault diagnosis applications.
Round 2
Reviewer 1 Report
The revised version of the manuscript is acceptable in present form.
Reviewer 2 Report
The authors have improved their contribution and have considered the reviewers' comments.